# Diffusion Mediates Molecular Transport through the Perivascular Space in the Brain

**DOI:** 10.3390/ijms25052480

**Published:** 2024-02-20

**Authors:** Marie Tanaka, Yoko Hirayoshi, Shinobu Minatani, Itsuki Hasegawa, Yoshiaki Itoh

**Affiliations:** Department of Neurology, Osaka City University Graduate School of Medicine, Osaka 545-8585, Japan

**Keywords:** glymphatic system, dextran, two-photon microscopy, cerebral cortex, penetrating vessel

## Abstract

The perivascular space has been proposed as a clearance pathway for degradation products in the brain, including amyloid β, the accumulation of which may induce Alzheimer’s disease. Live images were acquired using a two-photon microscope through a closed cranial window in mice. In topical application experiments, the dynamics of FITC-dextran were evaluated from 30 to 150 min after the application and closure of the window. In continuous injection experiments, image acquisition began before the continuous injection of FITC-dextran. The transport of dextran molecules of different sizes was evaluated. In topical application experiments, circumferential accumulation around the penetrating arteries, veins, and capillaries was observed, even at the beginning of the observation period. No further increases were detected. In continuous injection experiments, a time-dependent increase in the fluorescence intensity was observed around the penetrating arteries and veins. Lower-molecular-weight dextran was transported more rapidly than higher-molecular-weight dextran, especially around the arteries. The largest dextran molecules were not transported significantly during the observation period. The size-dependent transport of dextran observed in the present study strongly suggests that diffusion is the main mechanism mediating substance transport in the perivascular space.

## 1. Introduction

The aberrant accumulation of amyloid β (Aβ) in the cerebral cortex is regarded as the first step in the development of Alzheimer’s disease (AD), followed by tau accumulation [1,2,3]. The polymerization of Aβ forms protofibril, which is now experimentally proven to be the most toxic substance to neurons [4]. The elimination of Aβ in the brain is an important target of anti-Aβ antibody therapy, but the details of the clearance of degradation products are still unknown [5]. Recently, the “glymphatic system” was proposed as a lymphatic system in the brain that excretes large molecules from the brain parenchyma into the cerebrospinal fluid [6]. Supporting this model, Iliff et al. reported that fluorescently or radioactively labeled Aβ1-40 injected into the striatum can be observed around capillaries and draining veins. Aβ clearance was reduced in aquaporin 4 (AQP4) null mice, suggesting that perivascular flow may be involved in the clearance function [6,7,8,9,10]. However, the physiological role and detailed mechanisms of this system are still hypothetical, lacking dynamic data on “in vivo flow”. Our previous study confirmed, for the first time, that Aβ can be transported from the cortical surface to the deeper parenchyma through the perivascular space, although there is a difference in the transportation speed between oligomers and fibrils [11]. In the narrow perivascular space, large Aβ fibrils may be easily trapped between the vascular wall and parenchyma.

Dextran is a hydrophilic substrate with excellent biocompatibility that is often used for the experimental assessment of diffusion, microcirculation (blood flow) and membrane permeability [12]. It does not spontaneously polymerize and is mostly eliminated from the kidney without biodegradation. In this study, different sizes of FITC-labeled dextran were used to evaluate the transport capability of the perivascular space in mice using two-photon microscopy.

## 2. Results

### 2.1. Topical Application of Dextran

In the topical application experiments, 40 kD dextran was already observed circumferentially around blood vessels at the beginning of the observation period, 30 min after dextran administration (Figure 1A). The fluorescence intensity and number of arteries, veins, and capillaries with dextran accumulation did not significantly change for 120 min at any depth, suggesting that dextran accumulation had already reached a plateau within the initial 30 min (Figure 1B–D).

### 2.2. Continuous Infusion of Dextran over the Cortex

The transportation of dextran in the perivascular space surrounding the penetrating arteries (Figure 2A–C and Appendix A) and veins (Figure 2D–F and Appendix A) was observed as early as 10 min after the initiation of continuous infusion over the cerebral cortex. The intensity of dextran fluorescence increased gradually over the course of 30 min. The changes were prominent with 40 kD dextran (Figure 2A,D) and mild with 110 kD dextran (Figure 2B,E), but not noticeable with 2000 kD dextran (Figure 2C,F).

Most noticeably, the artery with its perivascular space changed its size and shape periodically, whereas the vein and its perivascular space remained still (Appendix A). The gradual increase in the dextran concentration was observed similarly in both the artery and vein, suggesting that vasomotion is not the main driving force of dextran transportation. The perivascular space was continuously observed from the penetrating artery to its branching arterioles (Appendix A).

Statistical analysis revealed that dextran significantly accumulated in a time-dependent manner for 30 min in the perivascular space around the penetrating arteries (*p* < 0.01, Figure 3A–C) and veins (*p* < 0.01, Figure 3D–F). A significant increase was observed with 40 kD and 110 kD dextran, but not with 2000 kD dextran. The increased accumulation of 40 kD dextran occurred faster than that of 110 kD dextran. These changes were commonly observed at all depths, although they were slower and less intense at deeper depths.

Similarly, time-dependent transportation of dextran was observed in the brain parenchyma (*p* < 0.01, Figure 4A–C). The size-dependent increase in the FITC-dextran intensity was statistically significant at a 60 μm depth, with the 40 kD dextran intensity increasing the fastest (*p* < 0.05, Figure 4A).

## 3. Discussion

In the present study, we demonstrated, for the first time, that dextran in the cerebral cortex can be transported through the perivascular space surrounding the penetrating arteries and veins to the parenchymal capillaries. The increase in the FITC intensity of smaller dextran molecules was faster than that of larger molecules, suggesting that diffusion is a mechanism of transportation.

In general, the mean-square travel distance of a particle diffusing in one dimension (*x*) is given by the Einstein–Smoluchowski equation:$$\bar{x^{2}}=2Dt$$
where *D* is the diffusion coefficient of the molecule and *t* is the length of time of the molecule [13]. When the molecule is approximated by a sphere of radius *r*, *D* is expressed by the Stokes–Einstein equation:*D* = *k_B_*
*T*/6*πrη*_0_
where *k_B_* is the Boltzmann constant, *η*_0_ is the solvent viscosity, and *T* is the absolute temperature. In the present study, we used dextran, which is not spherical but may extend its branches long enough to keep it hydrophilic. As the apparent radius of the largest dextran in this experiment is very large, the diffusion coefficient is calculated to be negligibly small, resulting in its diffusion being undetectable.

The present study demonstrates that substances in the perivascular space may be transported to the surface cerebrospinal fluid (CSF) by diffusion (Figure 5A). It further suggests that degradation products including Aβ, excreted from the brain parenchyma into the perivascular space, may be transported to the subarachnoid space by diffusion (Figure 5B).

In the extracellular space of the brain, most substances are transported via diffusion or convection [14]. Conventionally, diffusive mechanisms are dominant in the brain parenchyma [15,16], whereas convective flow in the CSF may be dominant in the brain ventricles and subarachnoid space [14]. Recently, Illif et al. proposed a novel system of interstitial fluid movement in which CSF enters the brain through the arterial perivascular space, flows through the brain parenchyma, and leaves through the venous perivascular space [6]. Their results suggested convective, one-way flow. Some reports have also indicated that the same artery has two layers of flow in opposite directions: the smooth muscle layer and the perivascular layer [17,18,19]. In the present study, we proved that dextran of different sizes can be transported through both the arterial and venous perivascular spaces by diffusion. Although the transport of dextran through the periarterial space was greater than that through the perivenous space, a delay in the venous system was not observed, excluding the possibility of convective flow from the cortical subarachnoid space into the deeper parenchyma.

In a previous study, we demonstrated that Aβ monomer can be quickly transported through the perivascular space, whereas polymerized Aβ can be trapped at the vascular wall [11]. These findings, together with those of the present study, suggest that Aβ monomer excreted into the perivascular space can be transported to the subarachnoid space by diffusion based on the concentration gradient, which decreases toward the cortical surface (Figure 5B). The dysfunctional removal of Aβ through the perivascular space is widely accepted as one of the mechanisms involved in the pathogenesis of AD [10]. To our knowledge, this is the first study to demonstrate that diffusion is the main force for excretion.

Clinically, the accumulation of Aβ in the vascular wall is known to cause cerebral amyloid angiopathy in the cerebral arteries and, to a lesser extent, in the veins [20]. Antibody therapy against Aβ for AD, now in clinical use, is known to induce inflammation and hemorrhage by reacting with the Aβ accumulated in the vascular walls. The glymphatic hypothesis proposed by Illif et al. cannot explain the pathogenesis of cerebral amyloid angiopathy in the arterial wall; however, our data are compatible with this pathophysiology.

The limitations of the present study include the following: (1) The perivascular transportation of dextran from the parenchyma to the cortical surface was not tested. (2) Much slower convective flow may have been missed, which may be relevant to large molecules with a small concentration gradient in the perivascular space. (3) The functional dilatation/constriction of the artery as a driving force of perivascular fluid was not tested. Although none of these limitations affect the significance of this study, further research is warranted.

## 4. Materials and Methods

### 4.1. Animal Preparation

The Osaka City University Ethics Committee on Animal Resources approved all experimental protocols used in this study (Protocol #22027). Animal experiments were conducted according to the protocol and the ARRIVE criteria. Male C57BL/6 mice (CLEA Japan Inc., Tokyo, Japan) aged from 6 to 13 weeks were used in this study. The mice were maintained on a 12 h light/dark cycle with the humidity and temperature controlled at normal levels and allowed food (CLEA Rodent Diet CE-2) and water ad libitum.

In all experiments, the animals were kept anesthetized with 1.5 to 2.0% isoflurane inhalation. After incising the scalp and exposing the skull, a 4 mm diameter cranial window was installed using a dental drill. In the topical application experiments, 3.5 µL of FITC-dextran solution was dropped onto the surface of the brain before closing the window, and images were obtained with a two-photon microscope. In the continuous injection experiments, a cranial window with an inlet and an outlet was placed on the brain surface, dextran solution was continuously injected at 2.0 µL/min, and image acquisition was started before dextran injection. In all the experiments described below, animals were excluded from the experiments only when the preparation was not sufficient for observation. At least five animals were used for the experiments requiring statistical analysis.

### 4.2. Preparation of Dextran Solutions

FITC-Dextran (TdB Labs, Ultuna, Sweden) 40 kD and 110 kD particles were dissolved in phosphate-buffered saline to 100 µM, whereas 2000 kD particles were dissolved to 10 µM to avoid insolubility.

### 4.3. In Vivo Observation with Two-Photon Microscopy

The dynamics of FITC-dextran transportation were observed using a two-photon laser microscope (A1RMP+1080, Nikon, Tokyo, Japan) equipped with a pulse laser, Chameleon Vision II (Coherent, Santa Clara, CA, USA), with a pulse width of 140 fs and a repetition rate of 80 MHz. The mice were immobilized on a stage below the microscope under isoflurane anesthesia. Fluorescent images were obtained using green (center wavelength, 525 nm; bandwidth, 50 nm) and red (center wavelength, 575 nm; bandwidth, 25 nm) bandpass filters with an excitation wavelength of 920 nm.

In the topical application experiments, the intensity measurement of FITC bound to 40 kD dextran could start only 30 min after dextran administration to close the cranial window and to set up the animal under a microscope. Images were obtained at 30, 90, and 150 min after application. The number of dextran-positive arteries, veins, and capillaries was determined visually in a square of 509 µm on each side (*n* = 5 for each condition) at depths of 40, 100, and 140 µm from the cortical surface.

To elucidate the early transportation of dextran, images were acquired continually before and 10, 20, and 30 min after the initiation of dextran infusion over the cerebral cortex. The luorescent intensity of FITC-dextran was measured in the region of interest surrounding the penetrating arteries or veins at depths of 60, 100, and 120 µm. The intensity ratio was calculated as the reference intensity measured at the center of each vessel. To examine the size dependence of dextran transportation, 40 kD, 110 kD, and 2000 kD dextran particles labeled with FITC were infused (*n* = 5 for each condition). In addition, the intensity of the brain parenchyma was measured at the area without major vessels in a square of 509 µm on each side (*n* = 5 for each condition). The intensity ratio was calculated relative to the reference intensity measured before injection.

To visualize the vessels, 8 μL/g weight of 5 mM of sulforhodamine 101 (Sigma-Aldrich, Saint Louis, MO, USA) was intravenously administered.

### 4.4. Statistical Analysis

In the topical application experiment, changes in the number of FITC-dextran-positive arteries, veins, and capillaries at various depths were evaluated using the Friedman test and the non-parametric multiple comparisons were conducted using SPSS 2.0 (IBM, Tokyo, Japan). In the continuous application experiment, changes in the intensity of FITC-dextran in the perivascular space of the penetrating arteries or veins and in the parenchyma, measured with three different sizes of dextran (40 kD, 110 kD, and 2000 kD), were compared using two-way repeated measures analysis of variance after confirming normal distribution, followed by post-hoc analysis using Dunnett’s test (*n* = 5 for each condition). Significance was set at *p* < 0.05.

## 5. Conclusions

This is the first study to demonstrate in vivo that substances in the perivascular space can be transported by diffusion with little effect on convective flow. Aβ polymerization in the perivascular space may, therefore, impede its diffusion, prompting the accumulation of Aβ in the vascular wall.

## Figures and Tables

**Figure 1 ijms-25-02480-f001:**
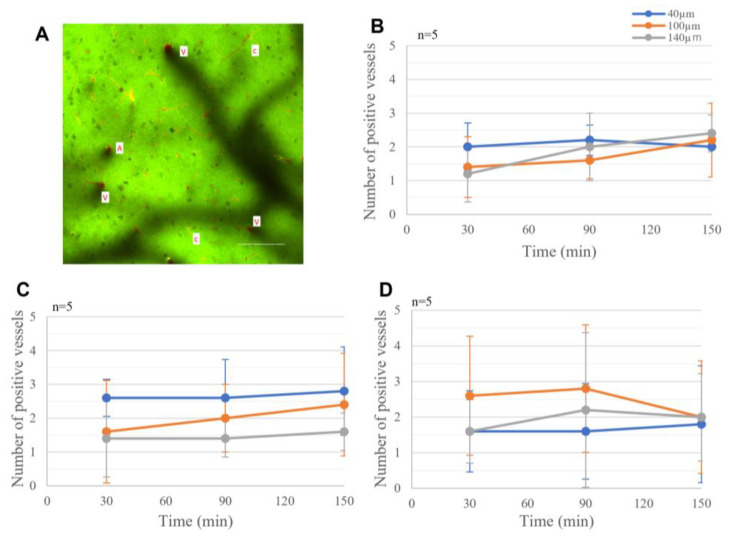
Dextran in the perivascular space in the topical application experiments. (**A**) Entire field of view at depth of 100 µm from the cortex 90 min after application reveals penetrating arteries (A), veins (V), and capillaries (c) with 40 kD FITC-dextran in the perivascular space (green) and intraluminal sulforhodamine 101 (red). Scale bar 100 µm. Average number of dextran-positive arteries (**B**), veins (**C**), and capillaries (**D**) in the observation field at depths of 40 µm, 100 µm, and 140 µm. No significant increase in the number of dextran-positive vessels was observed between 30 and 150 min.

**Figure 2 ijms-25-02480-f002:**
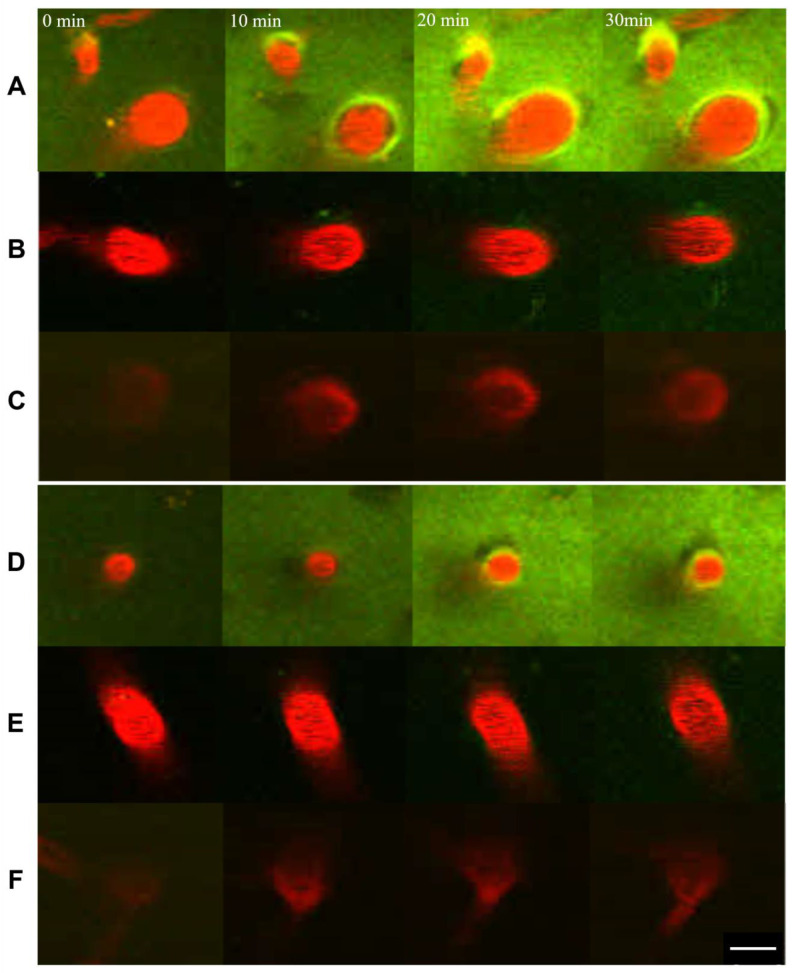
Transport of FITC-dextran (green) in the perivascular space of the penetrating vessels (red: sulforhodamine 101) at a 60 µm depth in the continuous injection experiment. Periarterial (**A**–**C**) and perivenous (**D**–**F**) FITC-dextran of 40 kD (**A**,**D**), 110 kD (**B**,**E**), and 2000 kD (**C**,**F**). The smaller molecules can be transported more rapidly in both periarterial and perivenous spaces. Parenchymal transport is also noticed with 40 kD dextran (**A**,**D**). Scale bar 10 µm.

**Figure 3 ijms-25-02480-f003:**
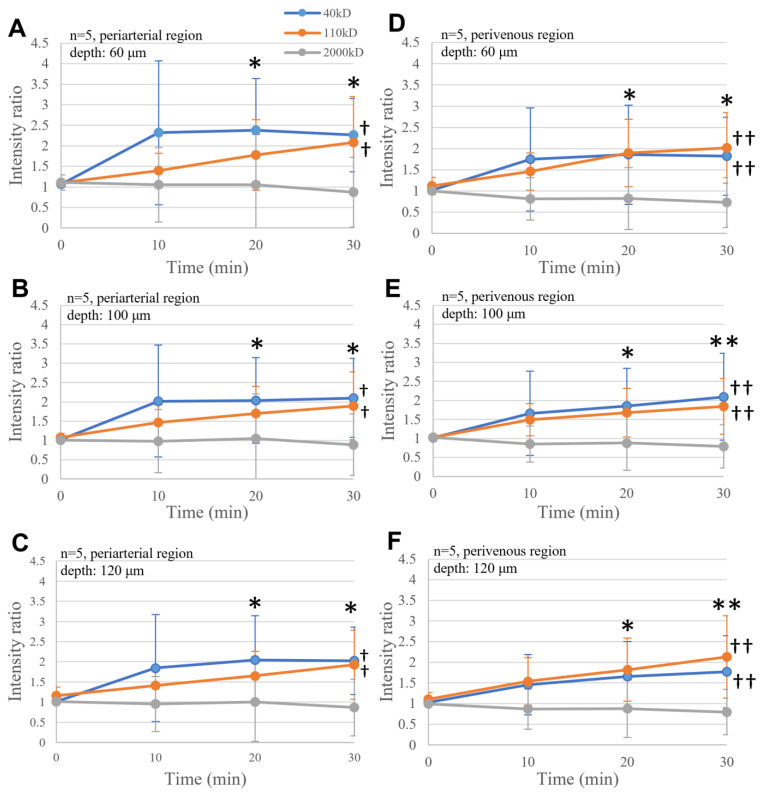
Periarterial (**A**–**C**) and perivenous (**D**–**F**) transport of FITC-dextran at depths of 60 µm (**A**,**D**), 100 µm (**B**,**E**), and 120 µm (**C**,**F**) in the continuous injection experiments. In the periarterial space, the increase in FITC-dextran fluorescence was significant with 40 kD and 110 kD dextran († *p* < 0.05), whereas a time-dependent increase was significant 20 and 30 min after the start of injection (* *p* < 0.05). The increase in 40 kD FITC-dextran fluorescence was fastest and was followed by that in 110 kD at 10 min. The intensity of 40 kD dextran reached a plateau at 20 min and was followed by 110 kD at 30 min. No increase in 2000 kD was noticed during the observation period. Similarly, in the perivenous space, the increase in dextran fluorescence was significant with 40 kD and 110 kD dextran particles (†† *p* < 0.01), whereas the time-dependent increase was significant at 20 and 30 min (* *p* < 0.05, ** *p* < 0.01).

**Figure 4 ijms-25-02480-f004:**
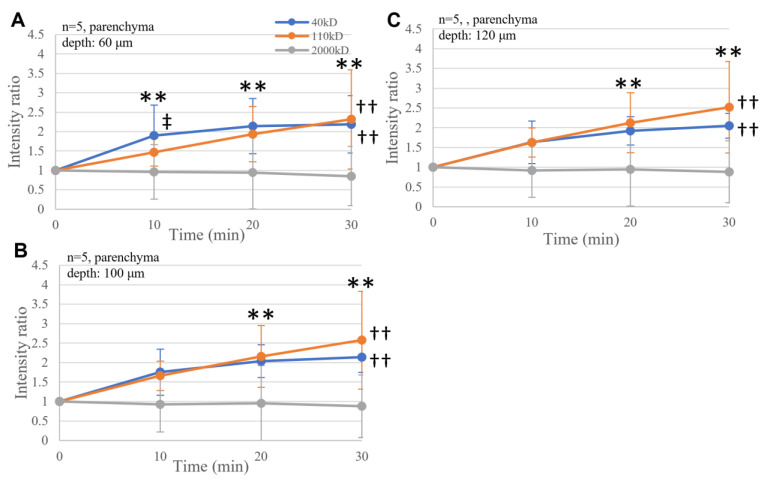
Transport of FITC-dextran through the brain parenchyma at depths of 60 µm (**A**), 100 µm (**B**), and 120 µm (**C**). Increased fluorescence was significant with 40 kD and 110 kD dextran (†† *p* < 0.01), whereas the time-dependent increase was significant at the times indicated (** *p* < 0.01). Noticeably, at a 60 µm depth, the transport was significantly size dependent, with smaller molecules moving faster (‡ *p* < 0.05).

**Figure 5 ijms-25-02480-f005:**
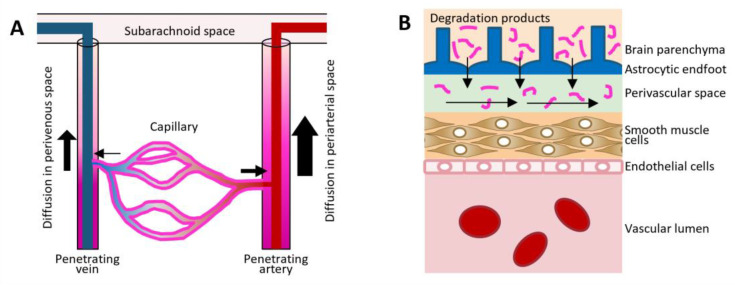
(**A**) Diffusion in the perivascular space may transport substances from the brain parenchyma through capillaries, arteries, and veins into the subarachnoid space. The present study showed that dextran can be reversibly transported from the subarachnoid space to the parenchyma. Diffusion is dependent on the concentration gradient, enabling degradation products to be excreted from the parenchyma. (**B**) Degradation products including Aβ, excreted into the perivascular space, may be transported through the space by diffusion. The oligomerization of Aβ can impede the diffusion, inducing the accumulation of Aβ along the arterial wall and resulting in cerebral amyloid angiopathy.

## Data Availability

Data is contained within the article and Appendix A.

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
