# Peer review of "Diffusion Mediates Molecular Transport through the Perivascular Space in the Brain"

_ijms, 2024, doi:10.3390/ijms25052480_

Round 1

Reviewer 1 Report

Comments and Suggestions for Authors

In this manuscript, titled "Diffusion mediates molecular transport through the perivascuar space in the brain", the transport of different sizes of dextran  molecules labeled with FITC  through the perivascuar space in the brain was evaluated by two-photon microscopy within 30 to 150mins range, Here is some comments

First, Why choose dextran as a model? is there any data or paper show that the dextran could mimic the amyloid β?

Second, Why choose the FITC as the labeling dye, Sometimes, the green fluorescent is difficult to identity.

Third, do you also try the monomer or small molecular weight dextran since the amyloid β has different conjugated size?

Comments on the Quality of English Language

Double check all spelling errors.

Author Response

Reviewer 1

We sincerely appreciate valuable comments by the Reviewer 1. Below is our response (in red ink) to the Reviewers’ comments (in black ink).

In this manuscript, titled "Diffusion mediates molecular transport through the perivascuar space in the brain", the transport of different sizes of dextran molecules labeled with FITC through the perivascuar space in the brain was evaluated by two-photon microscopy within 30 to 150mins range. Here is some comments.

First, why choose dextran as a model? is there any data or paper show that the dextran could mimic the amyloid β?

Dextran is a hydrophilic substrate with good biocompatibility used clinically for plasma substitute and experimentally for the assessment of diffusion, microcirculation (blood flow) and membrane permeability (Saxton, M.J. J Phys Chem B 2014, 118, 12805-17). It does not spontaneously polymerize and is mostly eliminated from the kidney without biodegradation. Wide range of molecular weight of dextran is commercially available. In these points, we chose dextran as most suitable tracer to evaluate the effect of molecular size on transportation in the perivascular space.

            Previously, we reported that polymerization of amyloid β reduce its transportation speed in the perivascular space. In the present study, different sizes of FITC-labeled dextran were used to evaluate the transport capability of the perivascular space.

            Following the comments, we modified the Introduction as below.

“Dextran is a hydrophilic substrate with excellent biocompatibility which is often used for experimental assessment of diffusion, microcirculation (blood flow) and membrane permeability [12]. It does not spontaneously polymerize and is mostly eliminated from the kidney without biodegradation.”

Second, Why choose the FITC as the labeling dye, Sometimes, the green fluorescent is difficult to identity.

In a small pilot study, we confirmed rhodamine-dextran can be transported through the perivascular space. We chose FITC-dextran for perivascular evaluation and rhodamine (SR101) for intraluminal visualization.

Third, do you also try the monomer or small molecular weight dextran since the amyloid β has different conjugated size?

We used 40kD, 110kD, and 2000KD dextran to test the different sizes of amyloid β polymer.

Reviewer 2 Report

Comments and Suggestions for Authors

The work, in its straightforward exposition, exhibits a robust impact and holds promise for future discoveries. I suggest that the authors elucidate the rationale behind the selection of mice at two different ages, specifically 6 an 13 weeks. Does this choice bear significance in terms of the experimental outcome? 

Another recommendation is to update some references that appear to be dated by more than 20 or 30 years.  

Author Response

Reviewer 2

We sincerely appreciate valuable comments by Reviewer 2. Below is our response (in red ink) to the Reviewers’ comments (in black ink).

The work, in its straightforward exposition, exhibits a robust impact and holds promise for future discoveries. I suggest that the authors elucidate the rationale behind the selection of mice at two different ages, specifically 6 and 13 weeks. Does this choice bear significance in terms of the experimental outcome?

We apologize for the error in the description of the method. We used mice aged from 6 to 13 weeks. Within this range, age did not affect the results. We corrected the sentence as below.

            “Male C57BL/6 mice (CLEA Japan Inc., Tokyo, Japan) aged from 6 to 13 weeks were used in this study.”

Another recommendation is to update some references that appear to be dated by more than 20 or 30 years. 

I believe that the classical reports by Braak in 1991 or Selkoe in 1991 are regarded as well-established, well-known, and cannot be substituted by the latter reports.

Reviewer 3 Report

Comments and Suggestions for Authors

The paper by Marie Tanaka et al, titled "Diffusion Mediates Molecular Transport through the Perivascular Space in the Brain," utilized a two-photon microscope to investigate the dynamics of FITC-dextran transport. The authors observed circumferential accumulation around penetrating arteries, veins, and capillaries from the beginning of the observation period. Continuous injection experiments revealed a time-dependent increase in fluorescence intensity around arteries and veins, with lower molecular weight dextran exhibiting faster transport, especially around arteries. Larger dextran molecules showed minimal transport, implying a size-dependent mechanism, predominantly diffusion, governing substance transport in the perivascular space.

While this research provides a comprehensive exploration of perivascular space transport, there are notable concerns that need addressing before considering acceptance.

Major concerns:

1.      The choice of dextran for the assay should be explicitly justified, detailing the rationale behind its selection. Additionally, the inclusion of another molecular species for control purposes is recommended.

2.      The authors need to provide explanations for the inability of large molecular weight dextran to undergo transport. A thorough discussion on this aspect is warranted.

3.      The conclusion references Ab polymerization affecting diffusion in the perivascular space, yet Ab is not investigated in this study. The authors should incorporate an experiment supporting this claim.

4.      The statistical results corresponding to Figure 2 should be included for clarity and rigor.

Minor concerns:

1.      Fig3 and Fig4 would benefit from added labels to enhance figure identification.

2.      Ensure consistency in the presentation of significance levels, making them uniform throughout the manuscript (e.g., either p < 5% or p < 0.05).

Author Response

Reviewer 3

We sincerely appreciate valuable comments by Reviewer 3. Below is our response (in red ink) to the Reviewers’ comments (in black ink).

The paper by Marie Tanaka et al, titled "Diffusion Mediates Molecular Transport through the Perivascular Space in the Brain," utilized a two-photon microscope to investigate the dynamics of FITC-dextran transport. The authors observed circumferential accumulation around penetrating arteries, veins, and capillaries from the beginning of the observation period. Continuous injection experiments revealed a time-dependent increase in fluorescence intensity around arteries and veins, with lower molecular weight dextran exhibiting faster transport, especially around arteries. Larger dextran molecules showed minimal transport, implying a size-dependent mechanism, predominantly diffusion, governing substance transport in the perivascular space.

While this research provides a comprehensive exploration of perivascular space transport, there are notable concerns that need addressing before considering acceptance.

Major concerns:

  1. The choice of dextran for the assay should be explicitly justified, detailing the rationale behind its selection. Additionally, the inclusion of another molecular species for control purposes is recommended.

Dextran is a hydrophilic substrate with good biocompatibility used clinically for plasma substitute and experimentally for the assessment of diffusion, microcirculation (blood flow) and membrane permeability (Saxton, M.J. J Phys Chem B 2014, 118, 12805-17). It does not spontaneously polymerize and is mostly eliminated from the kidney without biodegradation. Wide range of molecular weight of dextran is commercially available. In these points, we chose dextran as most suitable tracer to evaluate the effect of molecular size on transportation in the perivascular space.

            Previously, we reported that polymerization of amyloid β reduce its transportation speed in the perivascular space. In the present study, different sizes of FITC-labeled dextran were used to evaluate the transport capability of the perivascular space.

            Following the comments, we modified the Introduction as below.

“Dextran is a hydrophilic substrate with excellent biocompatibility which is often used for experimental assessment of diffusion, microcirculation (blood flow) and membrane permeability [12]. It does not spontaneously polymerize and is mostly eliminated from the kidney without biodegradation.”

  1. The authors need to provide explanations for the inability of large molecular weight dextran to undergo transport. A thorough discussion on this aspect is warranted.

Following the advice, we added further discussion in DISCUSSION as below.

“In general, the mean-square travel distance of a particle diffusing in one dimension (x) is given by the Einstein–Smoluchowski equation:

Average (x2 ) = 2Dt

where D is the diffusion coefficient of the molecule and t is the length of time the molecule. [13]. When the molecule is approximated by a sphere of radius r, D is expressed by the Stokes–Einstein equation:

D = kB T / 6πrη0

where kB is the Boltzmann constant, η0 is the solvent viscosity, and T is the absolute temperature. In the present study, we used dextran, which is not spherical but may extend its branches long enough to keep it hydrophilic. As the apparent radius of the largest dextran in this experiment is very large, diffusion coefficient is calculated to be negligibly small, resulting in its diffusion undetectable.”

  1. The conclusion references Ab polymerization affecting diffusion in the perivascular space, yet Ab is not investigated in this study. The authors should incorporate an experiment supporting this claim.

We have previously reported the effects of polymerization Aβ on the transportation speed (Hasegawa et al. reference 11). It was expressed in the INTRODUCTION as below.

“Our previous study confirmed for the first time that Aβ can be transported from the cortical surface to the deeper parenchyma through the perivascular space, although there is a difference in transportation speed between oligomers and fibrils [11].”

  1. The statistical results corresponding to Figure 2 should be included for clarity and rigor.

Figure 2 is representative samples of experiments illustrated in Figure 3 and 4. The results of statistical analysis including the data in Figure 2 were incorporated in Figure 3 and 4.

Minor concerns:

  1. Fig3 and Fig4 would benefit from added labels to enhance figure identification.

Following the comment, we added labels on each panel in Fig 3 and 4.

  1. Ensure consistency in the presentation of significance levels, making them uniform throughout the manuscript (e.g., either p < 5% or p < 0.05).

Following the comment, significance level is now uniformly expressed as p<0.05 or p<0.01 rather than 5% or 1%.

Round 2

Reviewer 1 Report

Comments and Suggestions for Authors

The author of this manuscript answer all my questions and modify the manuscript base on my comments.

Comments on the Quality of English Language

double check all the spelling error.